# Validation of Inflammatory Prognostic Biomarkers in Pleural Mesothelioma

**DOI:** 10.3390/cancers16010093

**Published:** 2023-12-24

**Authors:** Stephanie Iser, Sarah Hintermair, Alexander Varga, Ali Çelik, Muhammet Sayan, Aykut Kankoç, Nalan Akyürek, Betül Öğüt, Pietro Bertoglio, Enrico Capozzi, Piergiorgio Solli, Luigi Ventura, David Waller, Michael Weber, Elisabeth Stubenberger, Bahil Ghanim

**Affiliations:** 1Karl Landsteiner University of Health Sciences, Dr. Karl-Dorrek-Straße 30, 3500 Krems, Austria; 2Department of General and Thoracic Surgery, University Hospital Krems, Mitterweg 10, 3500 Krems, Austria; 3Department of Pathology, University Hospital Krems, 3500 Krems, Austria; 4Department of Thoracic Surgery, School of Medicine, Gazi University, Besevler, 06500 Ankara, Turkey; 5Department of Pathology, School of Medicine, Gazi University, Besevler, 06500 Ankara, Turkey; 6Division of Thoracic Surgery, IRCCS Azienda Ospedaliero Universitaria di Bologna, 40138 Bologna, Italy; 7Barts Thorax Centre, St Bartholomew’s Hospital, Barts Health NHS Trust, London EC1A 7BS, UK; 8Department of Thoracic Surgery, Sheffield Teaching Hospitals NHS Foundation Trust, Sheffield S10 2JF, UK; 9Division of Biostatistics and Data Science, Department of General Health Studies, Karl Landsteiner University of Health Sciences, Dr. Karl-Dorrek-Straße 30, 3500 Krems, Austria

**Keywords:** pleural mesothelioma, prognostic biomarkers, platelets, PLR, multimodal therapy, pleurectomy/decortication

## Abstract

**Simple Summary:**

Asbestos exposure is known as the main elicitor of pleural mesothelioma (PM) development. The pathology’s rarity, wide range of growth patterns, and devastating prognosis have hindered a standardized treatment to date. This study intended to determine possible prognosticators contributing to adjusting the treatment allocation. This initiated the analysis of the readily available biomarkers (from blood withdrawal) and clinical characteristics of 98 consecutive patients regarding their impact on overall survival (OS) in a retrospective and multicentered manner. Surgery (pleurectomy/decortication (P/D)), multimodal therapy (chemotherapy and surgery), a high hemoglobin level, a low platelet count, and a low platelet–lymphocyte ratio (PLR) were identified as favorable prognosticators. In multivariate analysis, histology, P/D, low C-reactive protein (CRP), and platelet levels were independent prognostic variables for this cohort. These validating results support further application of (lung-sparing) interventions and accompanying research on prognostic and predictive biomarkers.

**Abstract:**

Evoked from asbestos-induced inflammation, pleural mesothelioma represents a fatal diagnosis. Therapy ranges from nihilism to aggressive multimodality regimens. However, it is still unclear who ultimately benefits from which treatment. We aimed to re-challenge inflammatory-related biomarkers’ prognostic value in times of modern immune-oncology and lung-sparing surgery. The biomarkers (leukocytes, hemoglobin, platelets, neutrophils, lymphocytes, monocytes, neutrophil–lymphocyte ratio (NLR), lymphocyte–monocyte ratio (LMR), platelet–lymphocyte ratio (PLR), C-reactive protein (CRP)) and clinical characteristics (age, sex, histology, therapy) of 98 PM patients were correlated to overall survival (OS). The median OS was 19.4 months. Significant OS advantages (Log-Rank) were observed in multimodal treatment vs. others (26.1 vs. 7.2 months, *p* < 0.001), surgery (pleurectomy/decortication) vs. no surgery (25.5 vs. 3.8 months, *p* < 0.001), a high hemoglobin level (cut-off 12 g/dL, 15 vs. 24.2 months, *p* = 0.021), a low platelet count (cut-off 280 G/L, 26.1 vs. 11.7 months, *p* < 0.001), and a low PLR (cut-off 194.5, 25.5 vs. 12.3 months, *p* = 0.023). Histology (epithelioid vs. non-epithelioid, *p* = 0.002), surgery (*p* = 0.004), CRP (cut-off 1 mg/dL, *p* = 0.039), and platelets (*p* = 0.025) were identified as independent prognostic variables for this cohort in multivariate analysis (Cox regression, covariates: age, sex, histology, stage, CRP, platelets). Our data verified the previously shown prognostic role of systemic inflammatory parameters in patients treated with lung-sparing surgery within multimodality therapy.

## 1. Introduction

Pleural mesothelioma (PM) still carries a poor and heterogeneous prognosis, ranging from 8 to 30 months depending on the stage at diagnosis and the following treatment [1,2,3]. Asbestos exposure and the ensuing inflammatory process are critical players in its development [4]. PM treatment still needs to be standardized entirely since each histological subtype and tumor progression state demands a different approach, and several therapeutics are under investigation [4,5,6].

The uncertainty of successful patient-to-treatment allocation has urged and initiated biomarker research. This involves the identification of prognostic biomarkers signifying survival advantages and biomarkers predictive for therapy response [4]. Several inflammatory-related biomarkers have been identified and validated over the years. The blood levels of leukocytes, lymphocytes, monocytes, neutrophils, platelets, and albumin repeatedly showed to be of prognostic value in PM patients [7,8,9,10,11]. Ghanim et al. have even proven both C-reactive protein (2012) and fibrinogen (2014) to be of prognostic as well as predictive power in PM cases undergoing extrapleural pneumonectomy within multimodality therapy [12,13].

The immune system’s impact on the development and progression of PM holds promise for further biomarkers, especially within the frame of immunotherapy [7]. With immunotherapy and anti-angiogenic therapy on the rise, combined with lung-sparing surgery and its recently questioned role in PM, there is an urgent need for validated biomarkers. Patient cohorts that will benefit from novel treatment approaches should be identified to avoid unnecessary treatments and tailor personalized medicine for PM patients during lung-sparing surgery and modern (immune-)oncology.

## 2. Materials and Methods

### 2.1. Patients

In this retrospective multicenter study, data from 98 patients with a histologically proven PM diagnosis were pseudonymously collected and analyzed. The partaking departments are listed in Appendix A, alongside their contributing number of patients and the corresponding ethics committee, plus approval number.

All partaking departments conducted their data collection according to the ethical principles of the Declaration of Helsinki. Due to the retrospective approach of this study, no informed consent from patients was required. Throughout the study, the patients’ data were kept confidential.

The eligibility criteria included a histologically proven diagnosis of PM (epithelioid/biphasic/sarcomatoid), a comprehensive medical record documenting the PM treatments received at the participating hospital, and standard laboratory and clinical parameters.

A total of 65 patients received multimodal therapy (66.3%), which was defined as lung-sparing surgery in macroscopic radical intention together with chemo-/, radio-, or immunotherapy for this cohort. Further, 9 patients were treated with sole surgery (9.2%), 7 patients either received systemic chemotherapy, radiotherapy, or immunotherapy (7.1%), and 17 were treated with best supportive care (17.0%). Out of the patients who received cytoreductive surgical treatment (n = 74), 17 underwent a tumor-debulking (23%), and 57 received a macroscopic radical pleurectomy/decortication (=P/D; 77%), of which 29 (51%) received a P/D and 28 (49%) an extended P/D (=EPD). 

Sole surgery as a treatment for PM is not a therapeutic standard, but the inclusion of these patients was justified to gather a representative sample. Two of these nine patients passed away within one month after surgery, another two could not receive chemo- or radiotherapy due to comorbidities, and the remaining five were missing data on further treatments. 

The multimodally treated patients (n = 65) received the following therapies in addition to cytoreductive surgery: 34 (52%) patients received neoadjuvant chemotherapy, 4 (6%) received adjuvant chemotherapy, 17 (26%) had neoadjuvant chemotherapy and adjuvant radiation, 4 (6%) had adjuvant chemotherapy and radiation, 2 (3%) had neoadjuvant chemotherapy and immunotherapy, 2 (3%) received immunotherapy, and 2 (3%) received neoadjuvant chemotherapy, radiation, and immunotherapy. Seven (10.8%) patients eventually received best supportive care (BSC) as a result of therapy exhaustion.

### 2.2. Variables

The collected variables consisted of peripheral-blood-derived markers (leukocytes, hemoglobin, platelets, neutrophils, lymphocytes, monocytes, neutrophil–lymphocyte ratio (NLR), lymphocyte–monocyte ratio (LMR), platelet–lymphocyte ratio (PLR), C-reactive protein (CRP)) and clinical characteristics (gender, histology, stage, treatment), all of which were correlated to overall survival (OS). Solely the pre-interventional (pre-diagnostic/pre-surgery) withdrawn blood samples were considered for the statistical analyses, to prevent any intervention from significantly altering the inflammatory biomarkers’ levels.

The investigated biomarkers hemoglobin, platelets, leukocytes, neutrophils, monocytes, and lymphocytes were quantified via flow cytometry. The NLR was hence calculated with the absolute neutrophil count divided by the absolute lymphocyte count, the LMR with the absolute lymphocyte count divided by the absolute monocyte count, and the PLR with the absolute platelet count divided by the absolute lymphocyte count. The CRP levels were quantified through a latex agglutination test.

Clinical characteristics such as age, histology, gender, and therapy modality were available for all patients. The following blood-derived variables were collected in their entirety (n = 98): hemoglobin, platelets, neutrophils, lymphocytes, the NLR, and the PLR. In contrast, leukocytes (n = 7), monocytes (n = 8), the LMR (n = 8), and CRP levels (n = 10) were missing in some patients and could not be retrieved in retrospect.

### 2.3. Statistical Analyses

The statistical analyses were conducted using SPSS. *p*-values lower than 0.05 were counted as significant. The overall survival (OS) was defined from the date of diagnosis until the date of death or last follow-up. To calculate the OS, the Kaplan–Meier analysis was applied. The resulting curves were compared with a Log-Rank test. 

The data (biomarkers) were subdivided into below and above the calculated median to build same-sized groups for the categorical survival analyses and to further visualize the results using Kaplan–Meier graphs. Univariate survival analyses were conducted with categorical and metric data for comparison. On the one hand, this method was chosen to depict the differences in the distinct data processing because the median is not always the adequate cut-off and therefore does not yield clarifying results. On the other hand, calculations with metric data yield more accurate results because they depict whether a one-unit change in a certain biomarker would increase or decrease the risk of death.

Univariate and multivariate Cox regression were utilized to identify independent prognosticators and predictors of OS and calculate the interaction terms.

## 3. Results

As listed in Table 1, the cohort consisted of 25 female (25.5%) and 73 male (74.5%) patients with a histologically verified pleural mesothelioma diagnosis (n = 98, mean age: 64.9 ± 10.5 years; range: 42–88 years). Equally, 84 patients had epithelioid histology (85.7%), and 14 patients had non-epithelioid histology (14.3%), which consisted of 8 biphasic (8.2%) and 6 (6.1%) sarcomatoid cases.

### 3.1. Therapy Modalities, Including Surgery, Have an Undeniable Impact on OS

Besides biomarkers, the different therapy modalities had a significant impact on OS. Patients treated within the multimodality concept survived significantly longer than anyone else otherwise (surgery alone, C/R/I, BSC) treated (median OS: 26.1 vs. 11.7 vs. 9.2 vs. 3.2 months, *p* < 0.001, Table 1, Figure 1c). The median survival of sole surgery exceeded the C/R/I modality one’s. BSC, being ranked last in the median OS, met the expectations of palliative treatment. The cohort’s dichotomization according to surgery vs. no surgery revealed remarkable differences in OS. The different approaches, tumor debulking and P/D, did not yield significant differences in the median OS (tumor debulking: 24.3 months vs. P/D: 25.6 months) in contrast to patients without any surgical treatment (3.8 months, *p* < 0.001, Figure 1d).

### 3.2. The Prognostic Impact of Platelet Count and PLR

The total cohort’s median OS was 19.4 months (CI 12.1–26.7). The results of the univariate survival analysis, both via Log-Rank and Cox regression, proved the prognostic power of platelets and the PLR. The platelet count, dichotomized by its median, yielded significant survival advantages within the low-count cohort (26.1 vs. 11.7 months, *p* < 0.001, Table 1, Figure 1f). Similar results were found for the PLR: the below-median cohort survived significantly longer than the high-ratio cohort (25.5 vs. 12.3 months, *p* = 0.023, Table 1, Figure 1g). The platelet count and PLR were significant when calculated using metric and categorical data (Table 2).

The prognostic power of hemoglobin was proven both using metric data (HR 0.87, CI 0.77–0.99, *p* = 0.037, Table 2) and by category with 12 g/dL as a cut-off (*p* = 0.021, Table 1, Figure 1e), but not when dichotomized by the median (*p* = 0.250, Table 1). The age at diagnosis yielded significant results only with metric data (HR 1.03, CI 1.01–1.06, *p* = 0.041, Table 2) but not categorical (*p* = 0.057, Table 1, Figure 1a).

### 3.3. Histology, Monocytes, and CRP Show a Trend of Statistical Significance

The histological subtype could not prove its significant impact on survival for this cohort. In both univariate analyses, the results show a strong tendency toward statistical significance (*p* = 0.053, Table 1, Figure 1b; *p* = 0.056, Table 2). The monocyte count also yielded results near significance, but only when dichotomized by the median (p=0.051, Table 1; *p* = 0.053, Table 2), not with metric data (*p* = 0.260, Table 2). The CRP value proved its prognostic power with a solid trend of significance when calculated with metric data (*p* = 0.059, Table 2) but not by category (*p* = 0.437, Table 1; *p* = 0.438, Table 2).

Some variables could not be labeled as prognostic for this cohort. The patient’s sex did not reveal any impact on survival (*p* = 0.842, Table 1). Blood-derived biomarkers such as leukocytes, lymphocytes, neutrophils, and the corresponding ratios NLR and LMR remained insignificant throughout the analyses, not even displaying a trend toward statistical significance.

### 3.4. Multivariate Survival Analysis

To examine the independence of the platelet and CRP levels, a multivariate model adjusted for age, sex, histology, stage, and surgery vs. no surgery was created: see Table 3. In contrast to the preceding univariate survival analyses, histology and CRP (cut-off 1 mg/dL) proved their independent prognostic power in multivariate analysis, additionally to surgery vs. no surgery and platelet level. Non-epithelioid histology implied a 3.45-fold higher risk of death in opposition to the epithelioid subtype in this cohort (HR 3.45, 95% CI 1.6–7.4, *p* = 0.002). Whether patients received surgery or not repeatedly held strong statistical power, also within the multivariate model (HR 3.04, 95% CI 1.43–6.43, *p* = 0.004). Both the platelet (dichotomized by the median) and CRP levels (dichotomized by the cut-off 1 mg/dL) proved to independently influence the risk of earlier death (platelet level: HR 2.01, 95% CI 1.09–3.70, *p* = 0.025; CRP level: HR 1.76, 95% CI 1.03–3.00, *p* = 0.039).

### 3.5. Surgery vs. No Surgery Stratified by CRP Level

The prognostic impact of CRP was tested through additional stratification by surgery vs. no surgery: see Figure 2. In univariate analysis, apart from the analysis with metric data, which showed a trend, no significant association was found. In contrast, in multivariate analysis, CRP level was found to be an independent prognostic factor, which is the reason for its further stratification and interaction testing. Multivariate Cox regression yielded a significant interaction between CRP level (cut-off 1 mg/dL) and surgery vs. no surgery (see Figure 2 and Table 4). The survival curves, including the interaction term, indicate a complex relationship between CRP level, surgery, and prognosis.

On the one hand, a trend of a survival advantage with low CRP levels is clearly visible for both the operated and the non-operated patients (Figure 2). Nevertheless, there is no significant difference in the OS for low-level CRP/surgery patients as opposed to high-level CRP/surgery patients. The same applies to non-operated patients: the low-CRP/no-surgery curve suggests a significant difference in the OS overall compared to high-CRP/no-surgery. Yet, this distinction is primarily driven by a small subset of patients, concluding that CRP levels alone may not be considered prognostic in this analysis. The underlying cause is likely multifaceted. For instance, different additional therapies could be crucial to these results. After all, three patients received immunotherapy each in the low-CRP/surgery and high-CRP/surgery arm and two in the low-CRP/no-surgery arm. Still, this hypothesis was tossed because the majority had similar treatments (systemic chemotherapy and lung-sparing surgery), which leads us to the assumption that the intervention itself may be the reason for CRP’s lack of significance. 

On the other hand, the survival advantage for patients receiving surgery is indisputable. The previous comparison of surgery vs. no surgery revealed distinct results in OS (25.5 vs. 3.8 months), introducing the possible benefit of lung-sparing surgical procedures.

## 4. Discussion

The present study contributes to the growing literature on prognostic biomarkers and factors in PM management. Besides prognostic impact validation of therapy modality, hemoglobin, platelets, and the PLR, we determined the independent prognosticators, and (surgical) therapy itself being a crucial prognostic factor even after biomarker stratification. 

Among all the biomarkers, platelet count in particular proved its consistently reliable prognostic impact in our study. This was evidenced by its significant prognostic impact with both the categorical (*p* < 0.001) and metric data (*p* = 0.001) in univariate analyses and with confirmation of it as a prognosticator independent of age, sex, histology, stage, surgery vs. no surgery, and CRP in multivariate analysis (*p* = 0.025). Furthermore, we demonstrated that platelets were prognosticators for patients who received surgery (*p* = 0.003) and multimodal therapy (*p* = 0.018). These findings are not particularly surprising, as Ruffie et al. already described them several decades ago [14,15], just as they are described now [16,17,18]. It is common for cancer patients to have high platelet counts, a condition also referred to as reactive thrombocytosis as a side effect of neoplastic growth [19]. This is due to their promoting role in tumor development, growth, and spread [20]. On top of everything, platelets even affect the treatment efficacy, especially in chemo- and targeted therapies [20]. However, the research has not yet established a way to directly target platelet interactions with cancer cells without simultaneously harming their physiological function [20]. Until this is adequately exerted, identifying patients who could benefit from anti-thrombotic treatment may improve the prognosis for high-count patients [20]. Moreover, the supplementary (prognostic) investigation of mean platelet volume to assess the entirety of platelet activation and its effects has successfully been carried out for specific cancer types, and could also be helpful in PM research [21,22].

Similarly, but not to the same extent, the PLR has demonstrated its prognostic impact, although it is not considered an independent prognosticator. The ratio showed prognostic significance in both the univariate analyses, whether with categorical (*p* = 0.025) or metric data (*p* = 0.012). These results are unsurprising, given the significant impact of high platelet levels. In contrast, a low lymphocyte count in cancer patients decreases the lymphocyte-mediated tumor response, being associated with an unfavorable prognosis [23]. The prognostic power of the PLR has been described and validated for PM [24,25,26] and various other cancer types and cardiovascular diseases [23]. Therefore, this easily accessible biomarker, which reflects the relationship between the specific and non-specific immune system and thrombotic and inflammatory pathways, deserves continued attention in biomarker research.

Initially, when examined individually, neither histology nor CRP significantly affected survival in univariate analysis. However, their impact on survival became evident when we conducted multivariate analysis, identifying them as independent prognostic factors (histology: *p* = 0.002, CRP: *p* = 0.039). This observation aligns with the well-established understanding of the prognostic role of histological subtypes in PM. Historically, epithelioid, biphasic, and sarcomatoid subtypes have been recognized as important prognostic indicators, with epithelioid cases having the most favorable prognosis and sarcomatoid cases having the least favorable [27]. In our study, due to the relatively small sample sizes of eight biphasic and six sarcomatoid cases compared to the epithelioid group, we merged them into a non-epithelioid subcategory to enhance the statistical power. 

The cohort exhibits a similar pattern regarding the CRP levels. This biomarker has long been established as an inflammation parameter and is increasingly linked to malignancies [28]. Prospective studies suggest a heightened vulnerability to cancer among individuals with elevated serum CRP levels [28]. However, the prognostic impact of CRP levels for this cohort is controvertible. Except for a tendency toward significance in univariate analysis with metric data, the variable seemed irrelevant until its inclusion into the multivariate model yielded significant results.

Furthermore, even a significant interaction (*p* = 0.048) between CRP level (cut-off 1 mg/dL) and the variable surgery vs. no surgery was detected. CRP revealed a trend toward a negative prognostic impact for higher levels, but we could not deduce any beneficiary effect for (lung-sparing) surgically treated patients with low CRP values. On the contrary, Ghanim et al. demonstrated CRP’s prognostic and predictive role in PM already in 2012, but for patients treated predominantly with an extra-pleural pneumonectomy (EPP), the radical opposite [13]. Therefore, the extent of the intervention supposedly represents the deciding factor in whether serum CRP levels signify survival benefits.

Not to be overlooked is the impact the different treatment strategies had on the OS of this cohort. In univariate analysis, both multimodality vs. other therapies and surgery vs. no surgery yielded strong significant results (both *p* < 0.001), just as for surgery vs. no surgery in the multivariate analysis (*p* = 0.004). Aside from that, the cohort’s stratification according to treatment (surgery vs. no surgery) and biomarker level (CRP) pointed out the undeniably strong impact surgery has on OS (Figure 2). The multimodality concept stands out in its promising survival (median OS 26.1 months) compared to other therapies in this cohort, which supports its further application.

Surgery, in general, was a strong prognosticator, regardless of intent. Interestingly, the patients treated with tumor debulking shared quite a similar OS with the P/D-treated ones (median OS tumor debulking: 24.3 months vs. P/D: 25.4 months). Surgery alone is insufficient in treating PM patients. Still, it depicts the essential part of tumor burden reduction, as it is the major therapy component besides systemic chemotherapy in most guidelines [29,30]. This controverts the only recently published results of the Mesothelioma and Radical Surgery 2 (MARS 2) trial: sole systemic chemotherapy, as opposed to systemic chemotherapy + surgery (EPD), was described as more favorable regarding adverse events, quality of life, and risk of death, even if the median OS was similar [31]. Since, in MARS 2, the physicians only operated extensively (EPD), the comparison to this study regarding surgery as a prognostic variable is biased. Nonetheless, the immediate (surgical) reduction in tumor burden enhances the patient’s physical and psychological well-being. Whether surgery remains a relevant component in PM therapy is to be researched thoroughly and continuously, particularly in times of advances in systemic therapy, be they anti-angiogenics or immunotherapy.

Indeed, the retrospective nature of this study presented specific challenges, particularly regarding the treatment allocation and the availability of complete laboratory parameters. Owing to our collaboration with thoracic surgical centers, the number of patients who underwent surgery greatly exceeded the conservatively treated ones, which also included BSC patients. It is important to consider this limitation when examining the survival differences between the treatment options. Additionally, the recorded biomarker levels only provided snapshots of these parameters at specific time points. In light of these limitations, there is a growing need to leverage artificial intelligence software, particularly for gathering and analyzing data more comprehensively. For instance, exploring the trajectory of biomarkers and their post-operative behavior would be intriguing, shedding light on whether procedures like surgery can normalize aberrant biomarker levels. With a focus on biomarkers as prognostic and predictive factors, artificial intelligence could condense large amounts of data and provide novel insights. Moreover, integrating artificial intelligence into data collection and analysis can have global applications and transform our understanding of PM and its treatment. Due to this disease’s difficult therapeutic success, we must compare more heterogeneous cohorts involving every histological subtype and treatment regime in substantial numbers to derive clearer implications.

## 5. Conclusions

Overall, this study adds to the growing literature regarding prognostic biomarkers and factors in managing PM. Deciding whether to have (lung-sparing) surgery may not be as heavily dependent on inflammatory parameters as previously thought. Considering the promising OS of surgically treated patients, as opposed to the rest, the pursuit of its research is supported, especially in combination with biomarkers. This study’s significant results and tendencies may resemble other cohorts and be applied to prosper ideas for similar research questions. The prospective incorporation of artificial intelligence into these studies must clarify the role of biomarkers and the impact of diverse therapeutic strategies, potentially optimizing the treatment of PM patients and advancing their quality of life.

## Figures and Tables

**Figure 1 cancers-16-00093-f001:**
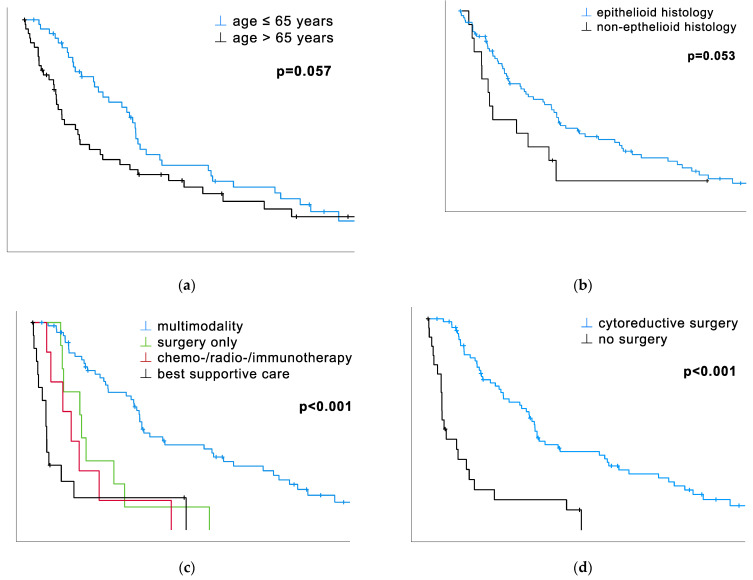
Prognostic factors for patients with PM with corresponding Kaplan–Meier survival curves calculated via Log-Rank. Variables are either dichotomized by their median, a chosen cut-off, or subcategories. See Table 1. (**a**) Patients aged younger than 65 years at diagnosis survived, with a trend of statistical significance, longer than the cohort above the median (25.6 vs. 12.3 months, *p* = 0.057). (**b**) The cohort with an epithelioid histological subtype survived longer than the non-epithelioid ones, as aforementioned, with a trend of statistical significance (24.2 vs. 7.8 months, *p* = 0.053). (**c**) Patients treated within the multimodality concept survived significantly longer than patients who received any other treatment (26.1 vs. 11.7 vs. 9.2 vs. 3.2 months, *p* < 0.001). (**d**) Patients who received cytoreductive surgery survived significantly longer than the ones who did not receive surgical therapy (25.5 vs. 3.8 months, *p* < 0.001). (**e**) Patients with hemoglobin levels above 12 g/dL survived significantly longer than those with levels below (24.2 vs. 15 months, *p* = 0.021). (**f**) Patients with a platelet level below 280 G/L survived significantly longer than the ones above the median (26.1 vs. 11.7 months, *p* < 0.001). (**g**) Patients with a PLR below the median of 195.6 survived significantly longer than patients above the median (25.5 vs. 12.3, *p* = 0.023).

**Figure 2 cancers-16-00093-f002:**
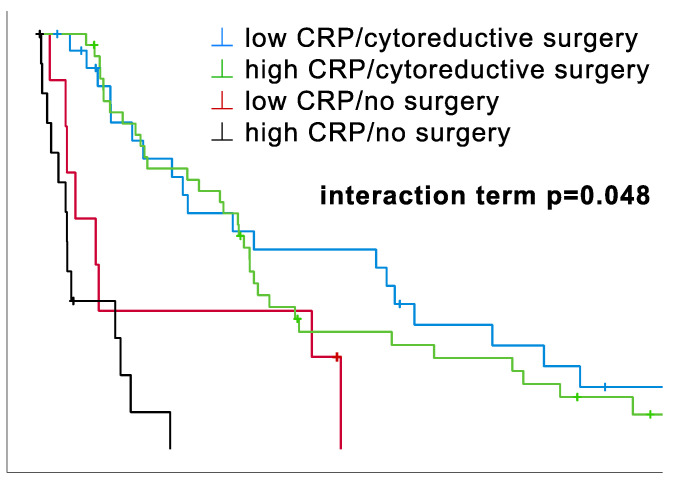
Impact of CRP (cut-off 1 mg/dL) levels on the prognostic effect of surgery vs. no surgery and multimodal therapy vs. other therapies. Kaplan–Meier survival curves highlight differences in overall survival (OS). The interaction term was determined through multivariate Cox regression, and p-values were calculated pairwise using the Log-Rank test. Survival subgroups categorized by CRP levels and surgery vs. no surgery. Median OS for surgically treated patients (low CRP: 26.1 (95% CI 0.0–61.9) months vs. high CRP: 24.9 (95% CI 21.2–28.6) months) exceeded that of non-surgically treated patients (low CRP: 6.8 (95% CI 0.0–14.1) months vs. high CRP: 3.3 (95% CI 3.0–3.7) months). For both surgically and non-surgically treated patients, differences in OS were insignificant, rejecting the statistical impact of CRP (cytoreductive surgery: *p* = 0.699, no surgery: *p* = 0.120). On the contrary, cytoreductive surgery revealed robust prognostic implications for both the low- and high-level CRP groups (low CRP: *p* = 0.009, high CRP: *p* < 0.001). Notably, the interaction term was significant (*p* = 0.048, Table 4).

**Table 1 cancers-16-00093-t001:** Univariate survival analysis via Log-Rank of the collected variables.

Variable	Cut-Off	n	% of the Whole Study Population	Median OS (Months)	95% CI	*p* *
Age at diagnosis	≤median 65.5 years >median	49 49	50.0 50.0	25.6 12.3	23.2–28.0 7.8–16.8	0.057
Sex	female male	25 73	25.5 74.5	25.6 18.1	11.2–40.0 9.7–26.5	0.842
Histology	epithelioid non-epithelioid	84 14	85.7 14.3	24.2 7.8	16.7–31.7 5.6–10.0	0.053
Treatment modality	multimodal surgery alone C/R/I BSC	65 9 7 17	66.3 9.2 7.1 17.3	26.1 11.7 9.2 3.2	23.1–29.2 10.5–13.0 4.1–14.4 1.8–4.5	**<0.001**
Surgery vs. none	cytoreductive surgery no surgery	74 24	75.5 24.5	25.5 3.8	23.0–28.0 2.3–5.4	**<0.001**
Hemoglobin	≤cut-off 12 g/dL >cut-off 12 g/dL ≤median 12.7 g/dL >median	36 62 49 49	36.7 63.3 50.0 50.0	15.0 24.2 16.1 23.6	9.2–20.7 15.8–32.6 4.5–27.8 13.7–33.5	**0.021** 0.250
Platelets	≤median 280 G/L >median	49 49	50.0 50.0	26.1 11.7	17.7–34.6 7.0–16.4	**<0.001**
Leukocytes	≤median 7.5 G/L >median	46 45	46.9 45.9	23.6 17.0	7.6–39.6 11.4–22.6	0.716
Neutrophils	≤median 5.2 G/L >median	49 49	50.0 50.0	24.9 16.1	16.9–32.9 9.0–23.3	0.794
Monocytes	≤median 0.7 G/L >median	45 45	45.9 45.9	25.6 12.8	21.1–30.1 7.0–18.6	0.051
Lymphocytes	≤median 1.5 G/L >median	49 49	50.0 50.0	15.0 24.3	7.0–22.9 16.6–32.0	0.602
NLR	≤median 3.3 >median	49 49	50.0 50.0	23.6 16.1	15.7–31.5 9.0–23.3	0.727
LMR	≤median 2.4 >median	45 45	45.9 45.9	15.0 24.3	8.7–21.2 15.2–33.4	0.601
PLR	≤median 195.6 >median	49 49	50.0 50.0	25.5 12.3	23.4–27.6 4.8–19.8	**0.023**
CRP	≤cut-off 1 mg/dL >cut-off 1 mg/dL ≤median 2 mg/dL >median	35 53 44 44	35.7 54.1 44.9 44.9	23.6 18.0 24.2 13.3	9.9–37.2 6.9–29.1 13.4–35.0 5.3–21.0	0.393 0.437

median OS = median overall survival; CI = confidence interval; *p* * = Log-Rank; C/R/I = chemo-, radio-, or immunotherapy; BSC = best supportive care; NLR = neutrophil–lymphocyte ratio; LMR = lymphocyte–monocyte ratio; PLR = platelet–lymphocyte ratio; CRP = C-reactive protein.

**Table 2 cancers-16-00093-t002:** Univariate Cox regression of the collected variables.

Variables	Cut-Off/Subgroup	HR	95% CI	*p* **
Age	metric median 65.5 years	1.03 1.54	1.01–1.06 0.98–2.40	**0.041** 0.059
Gender	female, male	0.95	0.56–1.60	0.842
Histology	epithelioid, non-epithelioid	1.84	0.98–3.44	0.056
Treatment modality	multimodal, surgery alone, C/R/I, BSC	1.83	1.51–2.23	**<0.001**
Surgery vs. none	cytoreductive surgery, no surgery	0.23	0.14–0.40	**<0.001**
Hemoglobin	metric cut-off 12 g/dL median 12.7 g/dL	0.87 0.57 0.77	0.77–0.99 0.35–0.92 0.49–1.21	**0.037** **0.022** 0.251
Platelets	metric median 280 G/L	1.003 2.39	1.001–1.004 1.49–3.84	**0.001** **<0.001**
Leukocytes	metric median 7.5 G/L	1.01 1.09	0.94–1.09 0.69–1.73	0.741 0.716
Neutrophils	metric median 5.2 G/L	1.01 1.06	0.93–1.09 0.68–1.66	0.836 0.794
Monocytes	metric median 0.7 G/L	1.20 1.59	0.94–1.29 0.99–2.54	0.260 0.053
Lymphocytes	metric median 1.5 G/L	0.92 0.89	0.66–1.29 0.57–1.39	0.628 0.603
NLR	metric median 3.3	0.99 1.08	0.92–1.07 0.69–1.70	0.836 0.728
LMR	metric median 2.4	0.82 0.89	0.64–1.05 0.56–1.41	0.122 0.611
PLR	metric median 195.6	1.002 1.68	1.00–1.004 1.07–2.64	**0.012** **0.025**
CRP	metric cut-off 1 mg/dL median 2 mg/dL	1.05 1.23 1.20	0.99–1.10 0.77–1.96 0.76–1.90	0.059 0.394 0.438

HR = hazard ratio, CI = confidence interval, *p* ** = univariate Cox regression, C/R/I = chemo-/radio-/immunotherapy, BSC = best supportive care, NLR = neutrophil–lymphocyte ratio, LMR = lymphocyte–monocyte ratio, PLR = platelet–lymphocyte ratio, CRP = C-reactive protein.

**Table 3 cancers-16-00093-t003:** Multivariate Cox regression model adjusted for age, sex, histology, stage, surgery vs. no surgery, platelet level, and CRP level. (n = 83, 85% of the study population).

Variables		Adjusted HR for Death	95% CI	*p* ***
Age at diagnosis	≤65.5 years >65.5 years	1 1.59	0.93–2.73	0.09
Sex	female male	1 1.02	0.55–1.90	0.96
Histology	epithelioid non-epithelioid	1 3.45	1.60–7.40	**0.002**
Stage	early (I, II) late (III, IV)	1 1.38	0.73–2.62	0.33
Surgery vs. no surgery	cytoreductive surgery no surgery	1 3.04	1.43–6.43	**0.004**
Platelet level	≤280 G/L >280 G/L	1 2.01	1.09–3.70	**0.025**
CRP level	≤1 mg/dL >1 mg/dL	1 1.76	1.03–3.00	**0.039**

HR = hazard ratio, CI = confidence interval, *p* *** = multivariate Cox regression, CRP = C-reactive protein.

**Table 4 cancers-16-00093-t004:** Multivariate Cox regression including the interaction term for CRP and cytoreductive surgery vs. no surgery.

Variable		HR	95% CI	*p* ***
CRP	≤1 mg/dL >1 mg/dL	1 3.42	1.33–8.75	**0.011**
Surgery vs. no surgery	no surgery cytoreductive surgery	1 0.39	0.17–0.90	**0.028**
CRP × surgery vs. no surgery	(=interaction term)	0.33	0.11–0.99	**0.048**

HR = hazard ratio, CI = confidence interval, *p* *** = multivariate Cox regression, CRP = C-reactive protein.

## Data Availability

All data and materials are available from the corresponding author upon reasonable request.

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
