# Peer review of "Validation of Inflammatory Prognostic Biomarkers in Pleural Mesothelioma"

_cancers, 2023, doi:10.3390/cancers16010093_

Round 1

Reviewer 1 Report

Comments and Suggestions for Authors

69: I think the wording "increasingly important role of lung-sparing surgery" should be ommited or changed as the role of surgery has to be further evaluated in prospective trial after MARS 2 results

85: partaking departments- can they be moved to supplement?

91: definition of multimodal therapy: immunotherapy is not included in that definiton, only surgery + chemotherapy +/- radiotherapy

91-97: majority of pts were treated with multimodality approach or were only treated by surgery: 66.3% + 9.2%. I think this really is an unusual group of selected pts , where on the other hand  only 7% were treated with systemic treatment only. This is an unusual selection of patients, in other centers accross the wolrd fewer pts are treated surgically. That makes those 2 groups (surgically treated and surgically not treated) uncomperable- to few pts in the other group. Also very short survival for those treated with systemic treatment only: 9.2mo 

321-323- I do not believe that CRP would ever lead our decision about offering patient P/D or not so the last prediction (in the last sentence) is too brave, I suggest ommiting it.

324-331:as stated above- a very small group of pts was treated with systemic therapy only and that is a great limitation of this trial, so you mostly compare surgically treated with BSC only and ofcourse, there is a huge survival difference

335: I do not think that 11.7 mo is a good srvival for only  surgically treated pts..would the live longer if treated with immunotherapy?

Comments on the Quality of English Language

very smooth , easily eading text, good English

Reviewer 2 Report

Comments and Suggestions for Authors

The manuscript “Validation of Inflammatory Prognostic Biomarkers in Malignant Pleural Mesothelioma” by Iser et al. is well written and of interest for scientists and clinicians working in this field.

I recommended the publication in Cancers, however, several points should be addressed.

- According to Sauter et al. (2021) “The 2021 WHO classification of Tumors of the Pleura: Advances since the 2015 Classification” the prefix malignant might be omitted from mesothelioma, because all mesothelioma cases are regarded as malignant.

- More information due to the sample collection should be presented, especially, the time-point of collection, i.e., the interval between sample collection and the begin of therapy.

- Information regarding the measurement of the biomarkers is missing.

- It remained unclear, why metric and categorical data were used in parallel. This should be explained in detail.

- The labels of the x- and y-axes are missing in Figure 1 and 2.

- It remained unclear why the Kaplan-Meier plot of hemoglobin is not presented in Figure 1, although the difference is significant using metric data (Table 3).
